# Modified Single-Walled Carbon Nanotube Membranes for the Elimination of Antibiotics from Water

**DOI:** 10.3390/membranes11090720

**Published:** 2021-09-21

**Authors:** Jana Gaálová, Mahdi Bourassi, Karel Soukup, Tereza Trávníčková, Daniel Bouša, Swati Sundararajan, Olga Losada, Roni Kasher, Karel Friess, Zdeněk Sofer

**Affiliations:** 1Institute of Chemical Process Fundamentals of the CAS, v.v.i., Rozvojova 135, 165 00 Prague, Czech Republic; bourassi@icpf.cas.cz (M.B.); soukup@icpf.cas.cz (K.S.); travnickovat@icpf.cas.cz (T.T.); 2Institute for Environmental Studies, Faculty of Science, Charles University, Benátská 2, 128 01 Prague, Czech Republic; 3IC2MP UMR 7285 CNRS, University of Poitiers, 4 rue Michel Brunet, CEDEX 9, 86022 Poitiers, France; 4Departments of Inorganic Chemistry, University of Chemistry and Technology Prague, Technická 5, 166 28 Prague 6, Czech Republic; Daniel.Bousa@vscht.cz (D.B.); zdenek.sofer@vscht.cz (Z.S.); 5The Department of Desalination & Water Treatment, Ben-Gurion University of the Negev, Beer-Sheva P.O. Box 653, Israel; swati14.ssr@gmail.com (S.S.); kasher@exchange.bgu.ac.il (R.K.); 6Departments of Physical Chemistry, University of Chemistry and Technology Prague, Technická 5, 166 28 Prague 6, Czech Republic; olgalosada1998@gmail.com (O.L.); karel.friess@vscht.cz (K.F.)

**Keywords:** carbon nanotube membranes, polymer, antibiotics, pertraction

## Abstract

The hydrophilic and hydrophobic single-walled carbon nanotube membranes were prepared and progressively applied in sorption, filtration, and pertraction experiments with the aim of eliminating three antibiotics—tetracycline, sulfamethoxazole, and trimethoprim—as a single pollutant or as a mixture. The addition of SiO_2_ to the single-walled carbon nanotubes allowed a transparent study of the influence of porosity on the separation processes. The mild oxidation, increasing hydrophilicity, and reactivity of the single-walled carbon nanotube membranes with the pollutants were suitable for the filtration and sorption process, while non-oxidized materials with a hydrophobic layer were more appropriate for pertraction. The total pore volume increased with an increasing amount of SiO_2_ (from 743 to 1218 mm^3^/g) in the hydrophilic membranes. The hydrophobic layer completely covered the carbon nanotubes and SiO_2_ nanoparticles and provided significantly different membrane surface interactions with the antibiotics. Single-walled carbon nanotubes adsorbed the initial amount of antibiotics in less than 5 h. A time of 2.3 s was sufficient for the filtration of 98.8% of sulfamethoxazole, 95.5% of trimethoprim, and 87.0% of tetracycline. The thicker membranes demonstrate a higher adsorption capacity. However, the pertraction was slower than filtration, leading to total elimination of antibiotics (e.g., 3 days for tetracycline). The diffusion coefficient of the antibiotics varies between 0.7–2.7 × 10^−10^, depending on the addition of SiO_2_ in perfect agreement with the findings of the textural analysis and scanning electron microscopy observations. Similar to filtration, tetracycline is retained by the membranes more than sulfamethoxazole and trimethoprim.

## 1. Introduction

Recently, emerging contaminants (ECs) have become a major threats to global water quality. Researchers and environmentalists have given serious attention to highlighting the sources of water contamination and proposing solutions [1,2]. A specific subcategory of ECs [3] represents pharmaceuticals. Compounds, such as antibiotics (ATBs), β-blocker hormones, blood lipid regulators, analgesics, and anti-inflammatory or cytostatic drugs, are widely used in daily life [4,5,6]. Industry, agriculture, hospitals, and households, through their water usage, release most of them [1,4,6,7]. Even though most of this water goes through wastewater treatment plants (WWTP), or sludge treatment plants, a portion of these pharmaceuticals remain in the water effluent of the treatment plants. The effluent is usually released to environmental surface waters (streams, rivers, lakes, reservoirs, and wetlands) [8,9] where the ATBs can be detected [10,11,12,13].

During the 20th century, ATBs underwent a tremendous evolution as crucial agents against microbial and fungal infection for humans, farms, and aquacultures. Yang et al. discussed ATBs as pharmaceuticals to protect humans and animals from disease and infection caused by bacteria [14]. According to Moffat et al., most of the dose is eliminated from the body within 24 h. However, less than 10% of this dose is excreted unchanged (the chemical molecule is sent directly to the treatment station) [15]. Sulfamethoxazole (SM), trimethoprim (TMP), and tetracycline (TET) were chosen as model ATBs in the present work.

More than seventy billion capsules of antibiotics were consumed during the year 2010. TMP and TET were classified fifth and sixth most consumed antibiotics in the world, respectively [16]. TET-based structures are polycyclic naphthacene carboxamide. They are used to treat malaria, cholera, or acne [17]. The principal use of TMP is to cure bladder infections. The intervention mechanism is via blocking folate metabolism [18]. SM, an ATB with a sulfonamide-based structure, is commonly used to cure bronchitis, urinary infections, irritations, etc. The activity mechanism is via inhibiting folic acid synthesis of the target bacteria, similar to other ATBs [18].

Despite conventional treatments, ATBs can persist in the effluents and resist the treatment of sewage treatment plants [19,20,21,22]. After being released to the surface waters, they can affect microorganisms, animals, plants, or algae living in the environment. Laxminarayan et al. tried to highlight global solutions to face ATB resistance [23]. Some other studies reported the toxicity of SM, TMP, TET to be 562.5, 100, and 182 mg/L for fish after 96 h exposure; 0.31, 0.15, and 25.5 mg/L for algae after 72 h of contact; and 74.2, 165.1, and NA mg/L for bacteria after 5 min exposure, respectively [24,25]. Moreover, diluted concentrations of ATBs improve the immunity of microbes, and the ATBs become ineffective. Studies from Rizzo [26] and Camargo [27] claim the advantage of the ATB treatment based on microbial activities. However, this can lead to a mutagenic reaction between the bacteria and the ATBs that develops and propagates genes across other bacteria strains. For example, TMP and SM are the most used ATBs for the inhibition of bacterial growth. The bacteria have gained some resistance toward SM; therefore, the two ATBs are used in synergy to overcome microbial immunity [28].

On the other hand, TET is also a widely used ATB for veterinary and human treatments. Once released into the environment, TET, as other ATBs, perturbs the ecosystem by inhibiting the growth of aquatic microorganisms and gathers via food chains until it reaches humans [29]. These ATBs cause different diseases, affecting the endocrine and nervous systems [10,29,30]. Regarding all the effects that these ATBs cause, scientists must propose efficient technologies for removing ATBs from the environment.

Conventional wastewater treatments can remove a considerable amount of ATBs. The removal mechanism is mainly by hydrolysis, adsorption, and degradation. However, these technologies strongly depend on the type of ATB and seasonal conditions [31]. It has been reported that WWTP in Italy could remove 71% of SM during the summer but only 17% during the winter. Moreover, the occurrence of ATBs in WWTP improves microbial resistance [31]. The resistance of Escherichia coli living in the effluent of WWTP toward different ATBs ranged from 26–100% [32]. Conventional treatments are insufficient for ATB elimination [8,9,13,20,30]. Pharmaceutical loaded wastewater depollution should be achieved from the source using advanced separation methods [19,33]. Several technologies have been proposed for pharmaceutical wastewater. Advanced oxidation processes are one of the new technologies still in development. Wet air oxidation [34,35], electrochemical, electrochemical photocatalytic, radiation assisted catalytic reaction, catalytic wet peroxide oxidation, and other oxidation methods all rely on generating hydroxyl, superoxide, and hydroperoxyl radicals, employing different chemical agents and activation energy sources under defined temperature and pressure conditions. The challenge is to reach the total mineralization of the pollutant and avoid toxic by-products. Other processes have recently been applied to full-scale wastewater treatment, e.g., adsorption on a different high surface and porous materials, such as clays, biochar [36,37], porous inorganic structures, activated carbon and carbon nanotubes (CNT), or graphene derivate. Some adsorption materials have shown a high-performance capacity even though they are limited by the adsorption capacity. Membrane processes have also been applied on a full scale and are the most used technology for EC treatment thanks to different processes and a wide range of membrane materials, depending on the leading separation mechanism of the membrane and targeted pollutant [38,39,40]. Moreover, some membranes combine other properties together, such as sorption capacity, degradation ability, and antifouling properties [41]. 

Combining material properties for separation processes to overcome limitations and challenges is a trend. This strategy is similar to improving the anti-fouling properties by doping materials in membranes to enhance pollutant degradation [41]. The property will prevent the membrane from quickly fouling during the separation process. The exact property observed on physically and chemically activated carbon/carbon nanotubes can expound the probable decomposition mechanism of sulfamethoxazole and carbamazepine during sorption on physically and chemically activated carbons. The study proved the presence of different interactions involved between ATBs and activated carbon responsible for ATB decomposition [42]. The sorption capacity of the CNTs is strongly related to the specific surface area. Others have compared the different CNT sorption capacities of Bisphenol A (BPA). Multi-walled carbon nanotubes (MWCNT), functionalized multi-walled carbon nanotubes, and single-walled carbon nanotubes (SWCNTs) were tested for their BPA sorption capacity. The CNT adsorption capacity followed this order: MWCNTs-COOH < MWCNTs < SWCNTs [43].

This work presents a novel approach for overcoming the adsorbent limitation of SWCNTs for ATB removal due to their sorption capacity. Application of the specific surface modification of different functionalized SWCNT membranes was successfully tested on three common ATBs (TET, TMP, and SM).

## 2. Experimental

### 2.1. Material

SWCNTs with a diameter of 1.6 ± 0.4 nm and a length of 5 µm were purchased from the OCSiAl company (Luxembourg). Tetraethyl orthosilicate (TEOS, 98%) were purchased from Sigma-Aldrich (St. Israel, MI, USA). Ammonia aqueous solution was purchased from Lachner, Czech Republic. Ethanol (p.a.) and dimethylformamide (DMF) were purchased from Penta (Slovakia, Czech Republic).

Monomer 2,2,3,4,4,4-hexafluorobutyl methacrylate (96%, HFBM) was purchased from Alfa-Aesar (Havrier, MI, USA). The photoinitiator, benzophenone, was purchased from Sigma-Aldrich, Israel. Dioxane was supplied by Biolab chemicals (HaNapah St, Ashkelon, Israe).

Tetracycline 98.0% (HPLC), sulfamethoxazole, and trimethoprim ≥ 98% (HPLC) were purchased from Sigma-Aldrich (Czech Republic). Deionised water was produced by an ultrapure water system (Simple Lab, Millipore S. A., Molsheim, France).

### 2.2. Membrane Preparation

#### 2.2.1. SWCNT Membranes 

Free-standing SWCNT membranes were prepared according to the procedure described below. First, 50 mg of bare unmodified SWCNTs was dispersed with 100 mL of DMF using high-energy shear-force milling using an UltraTurrax T18 set at 12,000 rpm for 30 min. The suspension was cooled with water circulating in the container shell and filtered by simple vacuum filtration. A filter with an active filtration diameter of 71 mm was mounted with a commercial polyester filtration (diameter 90 mm, thickness 0.4 µm). The remaining DMF solvent was rinsed out using 50 mL of ethanol. After that, the membrane was dried inside the filter equipment overnight and stored in a dark place for 24 h before further use. The prepared SWCNT membranes are listed in Table 1. The measurement of membrane thickness was done using micrometer on four points of membrane edge and two points in the middle of the membrane. The thicknesses are averages of those six measurements (standard deviation ±15%).

#### 2.2.2. Mildly Oxidized SWCNT Membranes

Mildly oxidized SWCNT membranes (SWCNT-MO membranes) were prepared according to the same procedure described in Section 2.2.1, except that mildly oxidized SWCNTs [44] were used for the fabrication process. Chemical oxidation of the SWCNTs was performed by a modified Hummer’s method, using potassium permanganate in an acidic environment. Compared to the classic Tour or Hummer’s oxidation procedure, a smaller amount of permanganate was used (1/8 of the original dose used in the Tour procedure). Oxidation was followed by a purification procedure performed by decantation and filtration until a neutral pH was achieved. The purified SWCNTs were then transferred to the distilled water and dispersed using high-energy shear-force milling (Ultra Turrax). A prepared stock water suspension of oxidized SWCNTs was stable and did not exhibit any sedimentation, even after 3 months. The prepared mildly oxidized SWCNT membranes are summarized in Table 2. 

Mildly oxidized SWCNTs were also used to prepare composite membranes containing black phosphorous particles (SWCNT-MO-BP). In the first step, 100, 200, or 300 mg of black phosphorus, prepared according to a procedure described elsewhere [45], was transferred into the duplicated glass vessel. DMF (100 mL), previously bubbled with argon for 4 h, was added to the vessel. The Ultra Turrax T18 dispersing tool was inserted into the liquid and sealed as much as possible with a plastic septum and parafilm to avoid the entrance of air. Argon (99.9%) was blown into the suspension during the entire milling time. High-speed milling using the Ultra Turrax (16,000 rpm) was used to exfoliate the black phosphorus (BP) bulk particles into the few-layer particles for 2 h. The glass vessel was cooled with water during the milling procedure. After 2 h, the milling was stopped, and the mildly oxidized SWCNTs (100, 200, or 300 mg) were added into the BP suspension and deagglomerated for an additional 1 h. At the end of the milling step, SWCNT-MO-BP membranes were prepared following the same procedure described in Section 2.2.1.

#### 2.2.3. Addition of SiO_2_ into SWCNT Membranes

The previously prepared membranes (from bare SWCNTs) were modified with SiO_2_ particles. The membranes were soaked for 24 h in a solution of TEOS (98 wt%) in order to provide enough time for the TEOS to wet the inner membrane surface. Then, the membranes were withdrawn from the TEOS solution and put onto a cotton napkin, which was gently pressed several times on both sides of the membrane to eliminate any surplus TEOS. Following that, the membranes saturated with TEOS were placed in an autoclave containing a mixture of ethanol (97 vol.%), ammonia (2 vol.%), and water (1 vol.%) such that the whole membrane was soaked in this solution. The sealed autoclave was placed in an oven at 125 °C for 12 h to provide enough time for the reaction between the TEOS and water to occur. A low concentration of water was carefully selected in order to achieve a slow hydrolysis reaction. At the end of the reaction, the membranes were withdrawn from the autoclave, dried at ambient temperature, and stored in the dark before further use. The SWCNT membranes enriched by SiO_2_ are listed in Table 3. The procedure of SiO_2_ precipitation inside of the membrane described above was done once for the SWCNT-200 ↘ SiO_2_ sample and three times for the SWCNT-200 **↗** SiO_2_ sample. 

#### 2.2.4. Modification of SWCNT Membranes by Graft Polymerization Using Hydrophobic Monomers

The SWCNT membranes were modified by UV-initiated graft polymerization using a perfluorinated (meth)acrylate HFBM monomer [46]. The modification was carried out as previously published [47] on one side of the membrane (Figure 1). The membranes were immersed in water prior to starting the modification followed by washing with a 4:1 ethanol–water mixture for 1 min. The membrane was fixed in a holder, and a benzophenone initiator with a concentration of 0.05 M was added to the upper part of the membrane for 1 min. It was again washed with the ethanol–water mixture (4:1) for 1 min to remove the unbound initiator. The monomer HFBM solution in dioxane with a concentration of 0.8 M was added to the UV record and pre-incubated for 1 min. Then, the record was placed in a UV chamber (Intelli-Ray 400, UV-tron International, West Springfield, MA, USA) for grafting under UV irradiation (mercury lamp; intensity = 33.056 mW/cm^2^ as measured by a UV light meter) for 10 min. The modified surface was washed once with dioxane and twice with the ethanol–water mixture (4:1) for 15 min under stirring. 

As a result of the graft polymerization, a hydrophobic top layer on the SWCNT composite membrane was created. Prepared composites with hydrophilic/hydrophobic layers on top are listed in Table 4.

### 2.3. Experimental Setup, Mode, and Evaluation Terminology

#### 2.3.1. Sorption Experiments 

The sorption of TET, SM, or TMP, as a unique pollutant or a mixture, using hydrophilic and modified membranes, was performed. The example of the sorption is illustrated in Figure 2. The experiments were performed in dark glass bottles. First, 70 mL of TET, SM, or TMP (or their mixture) in water (c_ATB_ = 200 mg/L) was put in contact with a membrane (at time 0) and was agitated at room temperature on a GLF 3005 rotator at 135 rpm. The active area of the membrane was 6 cm^2^. A 0.4 mL solution was sampled at regular time intervals and analyzed by high-performance liquid chromatography (HPLC). HPLC measurements were carried out on a Dionex 3000 UltiMate HPLC from Thermo Scientifics equipped with a binary pump, degasser, diode array detector (DAD), solvent tray, and an autosampler. The chromatographic column that was used was a Luna C18 (5 µm, 4.6 × 150 mm, Phenomenex). The detection wavelength was 254 nm. The HPLC method used the isocratic flow of a mobile phase composed of 20% acetonitrile and 80% ultra-pure water with 0.1% formic acid as a buffer. The temperature was 22 °C, and the flow rate was 0.3 mL/min for a total time of 20 min. Based on the previously performed calibration, the concentrations of ATBs were determined. The complete series of membranes were tested.

#### 2.3.2. Filtration Experiments 

The prepared SWCNT-MO membranes of three different thicknesses (118 µm, 193 µm, and 267 µm) and SWCNT-MO-BP membranes (125 µm, 156 µm, and 190 µm) were used for filtration of the ATB solutions in a classical dead-end setup with positive pressure as a filtration driving force. A smaller diameter membrane (47 mm) was cut from the previously prepared 90 mm membrane and placed in the ultrafiltration cell (Millipore). Then, 50 mL of the antibiotic solution (TET, TMP, and SM; 20 mg/L) was poured above the membrane, and pressurized air was used to create a positive pressure as a driving force for filtration. A constant air pressure and different membrane thicknesses led to various contact times of the ATB solution with the membrane. One membrane of a given thickness was used for filtration of all ATB solutions. Between each ATB solution, 20 mL of ethanol was filtered through the membrane in order to remove most of the previously adsorbed ATBs.

#### 2.3.3. Pertraction Experiments

Pertraction (PT) experiments were carried out in a closed, circular stainless steel cell (12 cm in length and 5.8 cm in diameter). The cell was divided into two chambers by a membrane fixed in a stainless steel disc. The scheme of the PT setup is shown in Figure 3. Experiments were performed at a constant temperature of 25 °C, which was maintained by recirculating ethanol through the double wall of the cell. 

Each membrane was cut to the chosen size using a round punch with a selected diameter and then fixed between two disc parts. The cell was then closed from both sides, and the chambers were filled with a stripping solution (deionised water) and feed solution (200 mg/L aqueous ATB) simultaneously so that the pressure was kept the same on both sides of the membrane. Both chambers were equipped with a magnetic glass stirrer, and constant stirring was achieved using external rotating magnets. The samples for HPLC analysis were extracted through the septa from the feed (0.4 mL) and the permeate side of the PT cell (0.4 mL) at regular time intervals by disposable sanitary syringes. At the beginning of the experiment (0, 1, 3 h), a higher frequency was followed by more extended time intervals depending on the separation rate. At first, the sampling was performed from the disrobing solution and then from the feed. Another needle was stuck through the septum while the sample was taken, preventing a pressure change in the chamber. The samples were analyzed by liquid chromatography, as described above. 

#### 2.3.4. Mathematical Model of the Diffusion Coefficients

The diffusion coefficients were evaluated using a simplified analytical model developed at the Institute of Chemical Process Fundamentals of the Czech Academy of Sciences, based on Fick’s 1st law. If mass transport occurs in the system solely by diffusion, the flow of the substance through the membrane surface can be expressed as: (1)$$\frac{1}{A}\frac{dn}{dt}=-D\frac{dc}{dx},$$
where n is the molar amount of ATBs, A is the active surface of a membrane, c is the concentration of ATBs, and D is the diffusion coefficient. The variables x and t are the length coordinate in the direction of diffusion and time, respectively.

In the case of a quasi-steady-state diffusion, the concentrations on the retentate and permeate side of the membrane are allowed to equilibrate. Assuming a constant retentate and permeate volume $V_{r}=V_{p}=V$ and an isotropic membrane material with a linear concentration profile across the membrane thickness, Equation (1) can be expressed as [49]: (2)$$\frac{dc_{p}}{dt}=\frac{DA}{V}\frac{\left(c_{r}-c_{p}\right)}{\delta },$$
(3)$$\frac{dc_{r}}{dt}=-\frac{DA}{V}\frac{\left(c_{r}-c_{p}\right)}{\delta },$$
where δ is the thickness of a membrane, and the subscripts, *r* and *p*, for the concentrations denote the retentate and permeate, respectively. 

Under the assumption that the amount of ATBs entering and leaving the membrane is the same,
(4)$$Vc_{p}+Vc_{r}=Vc_{0},$$
where $c_{0}$ is the initial concentration of the retentate. 

By combining Equations (2)–(4) and integrating the ordinary differential equations with the initial conditions (a) $c_{p}\left(t=0\right)=0$ and (b) $c_{r}\left(t=0\right)=c_{0}$, the relationship for calculating the diffusion coefficient can be obtained:(5)$$D=-\frac{V\delta }{2At}\ln\frac{\left(c_{r}-c_{p}\right)}{c_{0}},$$

### 2.4. Analysis and Characterization of the Membranes

The membrane morphology was determined using scanning electron microscopy (SEM) with a FEG electron source (Tescan Lyra dual-beam microscope). Elemental composition and mapping were performed using an energy-dispersive X-ray spectroscopy (EDS) analyzer (XMaxN) with a 20 mm^2^ SDD detector (Oxford instruments) and AZtecEnergy software. SEM and SEM–EDS measurements were carried out using a 10 kV electron beam.

Both nitrogen and krypton physisorption measurements at 77 K were performed using an automated volumetric gas adsorption analyzer ASAP 2020 (Micromeritics, Norcross, GA, USA). To guarantee the accuracy of the obtained adsorption isotherms, the highly pure nitrogen and krypton (grade of 99.999 vol.%) and helium (grade of 99.996 vol.% was used to determine the void volume typically performed before the analysis) were used. All samples were dried at 70 °C under a vacuum (<5 mbar) for 12 h prior to analysis. A self-made stainless steel measuring cell was used for the textural measurements due to the oversized diameter (<9 mm) of the tested membranes.

The specific surface area, *S_BET_,* was evaluated from the nitrogen or krypton adsorption isotherm in the range of the relative pressure corresponding to *p*/*p*_0_ = 0.05–0.25 for nitrogen and *p*/*p*_0_ = 0.05–0.20 for krypton using the standard Brunauer–Emmett–Teller (BET) approach [50]. The total specific pore volume was determined from the corresponding adsorption isotherm by converting the equilibrium amount adsorbed at a relative pressure of *p*/*p*_0_ = 0.99 into the liquid volume, assuming that the density of the adsorbate is equal to the bulk liquid density at saturation. The mesopore surface area, *S_meso_*, and the micropore volume, *V_micro_*, were determined by the *t*-plot method [51]. The mesopore size distribution was evaluated from the desorption branch of the nitrogen adsorption–desorption isotherm using the Barrett–Joyner–Halenda (BJH) method via the Roberts algorithm [52,53]. The carbon black statistical thickness (STSA) method for the thickness curve calculation was used for the *t*-plot and the mesopore size distribution evaluation.

The sessile drop water contact angle measurements were carried out using the OCA 20 (Dataphysics Products, Filderstadt, Germany). A drop volume of 2 µL was used, and the measurement indicates an average of six readings. Surface elemental composition by X-ray photoelectron spectroscopy (XPS) of the unmodified and modified membranes was performed using an Escalab 250 spectrometer (Thermo Fisher Scientific, Waltham, MA, USA).

The membranes, fresh and used, were analyzed by Fourier transform infrared (FTIR) spectroscopy. An FTIR spectrometer Avatar 360 (Nicolet) was used to measure the IR spectra of the samples in the range of 508 to 4000 cm^−1^ (resolution 1.93 cm^−1^, 200 scans, 1 s per scan). FTIR spectroscopy in the attenuated total reflection (ATR) mode was used to obtain spectra from the membrane pressed against a ZnSe crystal. 

## 3. Results

The SWCNT membranes were grafted on a top-layer using the hydrophobic hyper fluorinated monomer HFBM ATR. The grafting was performed using free-radical polymerization using a photo-initiator, benzophenone. Hydrophobic graft polymerization aims to obtain a combined hydrophilic/hydrophobic layer on top of the porous support membrane to achieve improved and advanced membrane properties.

### 3.1. Characterization of the Fresh Membranes

#### 3.1.1. Water-Drop Contact Angle with XPS

Free-standing The contact angle studies of the modified membranes were carried out to determine the hydrophobicity of the membranes, and the results are presented in Table 5. The contact angle of the unmodified SWCNT-200 membrane was found to be 63.5 ± 3.5°. The membrane grafted with HFBM showed a higher contact angle of 80.1 ± 4.4°, indicating a more hydrophobic, low energy surface, confirming a successful hydrophobic graft polymerization. The contact angle of the other modified membranes, namely SWCNT-200 ↘ SiO_2_ and SWCNT-200 ↗ SiO_2_, was impossible to achieve, as they showed a highly porous nature.

XPS was used to analyze the surface elemental composition of the membranes to understand the changes due to chemical functionalization. The results are shown in Table 5. All the modified membranes show the F1s peak at a binding energy of 688.5 eV, indicating successful grafting of the HFBM monomer onto the membrane surface. All the membranes exhibited C1s and O1s peaks at binding energies of 284.8 and 532.5 eV, respectively. The unmodified SWCNT-200 membrane contained 95.25% carbon and 4.75% oxygen. The presence of the C1s peak at 284.8 eV can be attributed to the sp^2^ hybridized graphitic carbon. SWCNT membranes with SiO_2_ show a high oxygen content and a low carbon content. The Si2p peak in SWCNT-200 ↗ SiO_2_ and SWCNT-200 ↘ SiO_2_ was observed to be at 104.4 eV along with the C1s and O1s peaks. 

#### 3.1.2. SEM Observations

All membranes were characterized by SEM (Figure 4). The membrane prepared from unmodified SWCNTs (SWCNT-200) exhibited a typical structure of entangled SWCNTs aggregated into bundles of multiple CNTs. The membrane prepared from mildly oxidized SWCNTs (SWCNT-MO-200) exhibited an almost identical structure to that of a CNT-bare membrane. However, the mildly oxidized CNTs form thicker bundles, most likely due to increased attraction forces between the CNTs caused by hydrogen bonding between oxygen functionalities. 

The membranes modified with SiO_2_ nanoparticles (SWCNT-200 ↘ SiO_2_ and SWCNT-200 ↗ SiO_2_) clearly show small particles of SiO_2_ deposited on the surface of the CNTs. The membrane SWCNT-200 ↗ SiO_2_ exhibits a structure with a significantly higher amount of SiO_2_ nanoparticles, which is in good agreement with the elemental composition obtained by EDS (Table 6). The purpose of SiO_2_ nanoparticle synthesis on the surface of the carbon nanotube was to alter the membrane pore structure and pore diameter in order to influence ATB pertraction. 

Furthermore, highly hydrophobic ATBs are more likely to penetrate through hydrophobic membranes, and thus the surface of the prepared membranes was coated with a grafted fluorinated polymer. This modification led to the formation of additional puff-like structures on the membrane surface, as shown in the SEM images for the Mod_SWCNT-200 ↘ SiO_2_ and Mod_SWCNT-200 ↗ SiO_2_ samples. Even though this hydrophobic layer does not form a uniform layer, it completely covers the CNTs and SiO_2_ nanoparticles and provides a significantly different membrane surface for interaction with the ATBs. On the other side, the membrane prepared out of SWCNTs (Mod_SWCNT-200) shows no puff-like structure, suggesting no bonding of the grafted fluorinated polymer on the CNTs did happen. However, this could be expected because the nature of both constituents is different (hydrophilic CNTs and the hydrophobic grafted fluorinated polymer). The chemical composition of the above-discussed membranes is shown in Table 6. Elemental composition of the membrane is the main factor for membrane–pollutant affinity, determined by the functional groups present on membrane surface. The elemental analysis helped us to keep track with modification steps by confirming the presence of SiO_2_ and HFBM monomer for modified membranes.

#### 3.1.3. Texture Analysis

Selected membranes were characterized by nitrogen or krypton at 77 K physisorption measurements. The results of the textural assessment of the measured membranes are summarized in Table 7. The SWCNT-MO-BP-400 sample showed a nitrogen-based BET surface area lower than 1 m^2^/g. For that reason, the specific surface area was calculated from the corresponding adsorption isotherm of krypton. 

The unmodified membranes, SWCNT-200, SWCNT-200 **↘** SiO_2_, and SWCNT-200 **↗** SiO_2_, reveal a higher magnitude of the specific surface area (*S_BET_*), the specific surface area of the mesopores (*S_meso_*), as well as the micropore volume (*V_micro_*) than the modified membranes (Table 7). However, this group of samples shows no significant differences between the *S_BET_* characteristics, somewhat corresponding to a measurement error than the different microstructure in terms of texture. On the other hand, the total pore volume increases with an increasing amount of SiO_2_ in the unmodified membranes (Figure 5a).

For the modified membranes, *S_BET_* decreases with increasing SiO_2_ content (Table 7). Contrary to the unmodified membranes, the total pore volume shows minor variations (Figure 5a) and a lower micropore volume (Table 7). The primary adsorption isotherms of nitrogen at 77 K are summarized in Figure 6. The observed hysteresis loop between the adsorption and desorption branches of the sorption isotherms is indicative of mesoporosity. Additionally, the specific surface area of the mesopores, *S_meso_*, is evaluated by the *t*-plot approach and given in Table 7. A significant volume of the mesopores in all studied membranes was confirmed. Furthermore, the steep increase in the equilibrium adsorbed amount (*a*) close to the saturation vapor pressure represents the pore filling of large meso- and even macropores (pores larger than 50 nm).

The pore size distribution (PSD) functions of the corresponding membranes are depicted in Figure 7. As can be seen, all tested samples revealed a unimodal distribution with a single highest value corresponding to the area of the mesopores. The determined maxima of a pore width for SWCNT-200, SWCNT-200 **↘** SiO_2_, and SWCNT-200 **↗** SiO_2_ membranes were found to be 23, 31, and 49 nm, respectively. On the other hand, the same influence of the SiO_2_ content on the PSD was not observed in a group of modified membranes (the frequency curve maxima at 30, 27, and 48 nm, corresponding to Mod_SWCNT-200, Mod_SWCNT-200 **↘** SiO_2_, and Mod_SWCNT-200 **↗** SiO_2_, were found).

### 3.2. Sorption Experiments

Table 8 lists the three representative antibiotics, SM, TMP, and TET, and their mixture, which were chosen for the set of 24 sorption experiments. The mildly oxidized SWCNT and SWCNT with black phosphorus (BP) were tested. The initial concentration was chosen based on the water solubility for TET: 331 mg/L, TMP: 397 mg/L, and SM: 379.6 mg/L.

The HPLC spectra of the mixture of ATBs before the sorption experiment and after 1 day using SWCNT-MO-200 are shown in Figure 8. 

SWCNTs are well known for their excellent sorption capacity. Mildly oxidized SWCNT membranes with or without BP adsorbed the entire amount of each ATB—TET, TMP, and SM—in less than 8 days (Figure 9). The membranes containing BP adsorbed slightly slower than without BP, except for SWCNT-MO-BP-100/300 and SWCNT-MO-BP-200/200 in the elimination of TET, which lasted just 1 h. The membranes without BP (SWCNT-MO-200, SWCNT-MO-400, and SWCNT-MO-600) that had different amounts of CNTs (200, 400, and 600 mg) and, thus, thicknesses (118, 193, and 267 µm) did not show any significant variations in the sorption process. All three membranes eliminated the entire amount of the three ATBs in less than 5 h. The same results were obtained in the treatment of the mixture of the three ATBs.

The membranes without BP were chosen for further development of separation materials for water treatment (WT) of ATBs. The primary sorption experiments ended with a clear water solution. In order to estimate the maximal sorption capacity of the membranes, the same amount of ATBs (c_ATB_ = 200 mg/L) with 40 times smaller membranes (15 mm^2^) was used for the new sorption experiments. Table 9 lists the weight of the adsorbed ATBs (SM, TMP, and TET) and the amount of substance per area of the membrane after saturation of the SWCNT membranes (SWCNT-MO-200, SWCNT-MO-400, and SWCNT-MO-600). No matter which ATB was eliminated, the sorption capacity increased with the increasing amount of CNTs in the membrane (200, 400, and 600 mg of CNTs before milling during the preparation of the SWCNT-MO-200, SWCNT-MO-400, and SWCNT-MO-600 membrane). In other words, the thicker membrane leads to a higher amount of ATBs being adsorbed.

Gathering the results from the first sorption (Figure 9) with the saturation tests (Table 9), it seems that ATBs are adsorbed similarly (Figure 9 on the right). However, the maximal amount of ATBs for each membrane varies (Table 9). The amount of adsorbed substances (in mol) is linked to the diameter of the spherical enclosure, *d*_SE_ (in Å), and it is influenced by the polar surface area (PSA (in Å^2^)). These parameters are written and illustrated in Figure 10. 

The amount of adsorbed substances (the moles of ATBs) per area of the membrane after saturation of mildly oxidized SWCNT membranes is very similar for SM and TMP (5.9, 11.8, and 15.8 moles of SM versus 6.9, 12.0, and 15.5 moles of TMP). These two antibiotics have *d*_SE_ and PSA parameters that are also very close (*d*_SE_ equals 12.0 vs. 13.5 Å and PSA equals 107 vs. 106 Å^2^, respectively). On the contrary, TET displays relatively elevated values for all three parameters (n of adsorbed TET per surface unit is 13.5, 19.1, and 22.5 moles depending on the membrane, *d*_SE_ equals 15.0 Å, and PSA is equivalent to 182 Å^2^). In particular, an elevated polar surface might be the reason for a TET reactivity with the mildly oxidized membranes.

### 3.3. Filtration Experiments

In the next step, a simple filtration experiment using a pressure filtration cell with three membranes of different thicknesses (118, 193, and 267 µm) was performed. Filtration of pure water was performed at a constant air pressure generated in the air compressor (up to 4.14 bar) in order to describe the system in terms of contact time and flow rate.

The influence of pressure on flow rate is relatively low (see SWCNT-MO-600 and -400 in Figure 11, upper part) if a sufficiently thick membrane was used. However, thinner membranes put a lot less resistance to the water flow, and a change in pressure significantly changes flow rates (at 4 bar, the flow rate is more than four times higher than at 1 bar). The calculated contact time obtained from the membranes thickness and flow rate at a given pressure is shown in Figure 11, bottom part.

Filtration of a solution containing an individual ATB was performed at 4.14 bar. Thus, a membrane with a thickness of 118 µm has a contact time of 2.3 s, a 193 µm membrane has a contact time of 6.7 s, and the thickest membrane (267 µm) has a contact time of 11.8 s. Even a short contact time, such as 2.3 s in the thinnest membrane (118 µm), was still sufficient for eliminating 98.8% of SM, 95.5% of TMP, and 87.0% of TET. As shown in Figure 12, thicker membranes (193 and 267 µm) exhibit very similar removal rates, indicating that the contact time of 2.3 s was sufficiently long to remove ATBs. However, thicker membranes contain a higher amount of SWCNTs and possess a higher adsorption capacity. On the other side, thinner membranes exhibit much higher flow rates, and thus they can be used for applications where there is a need for quick removal of ATBs from small volumes of contaminated water.

### 3.4. Pertraction Experiments

The modified membranes Mod_SWCNT-200, Mod_SWCNT-200 ↘ SiO_2_, and Mod_SWCNT-200 **↗** SiO_2_ underwent PT experiments to study the permeation of the ATB mixture (SM, TMP, and TET) through the hydrophobic membranes. The amount of ATBs (%) in the feed and permeate for the modified membranes as a function of time is visualized in Figure 13. 

Comparing the elimination of three ATBs from the feed, the increasing amount of SiO_2_ in the modified membranes enhances the removal. It can be linked to the pore size distribution of these membranes (Figure 7, on the right), as discussed in Section 3.1.3 Textural Analysis. The most visible difference in the feed is TET (Figure 13, in blue), while SM and TMP behave similarly (Figure 13, in orange and grey colors, respectively). The different behavior of the three ATBs was described above (Table 9 and Figure 10). Moreover, TET passes through the Mod_SWCNT-200 membrane before being reabsorbed into the membrane. Mod_SWCNT-200 ↘ SiO_2_ and Mod_SWCNT-200 **↗** SiO_2_ kept TET all along with the PT experiments. Only two ATBs—SM and TMP—can be seen in the permeate (Figure 13, on the left). These molecules react less with the membrane (their polar surface area is much lower than the one of TET) (Figure 10). 

The concentration in the feed and permeate in time and an example of the diffusion coefficient calculation using a mathematical model are illustrated in Figure 14.

The flux is an important parameter to characterize the efficiency of the transport through the membrane. The diffusion coefficients of SM, TMP, and TET transport through the selected membranes are listed in Table 10. The measured values vary between 0.7–2.7 × 10^−10^. Two values could not be measured due to the missing amount of TET in the permeate. 

Comparing the three membranes, regardless of which ATB was investigated, the diffusion coefficient follows the order: Mod_SWCNT-200 < Mod_SWCNT-200 ↘ SiO_2_ < Mod_SWCNT-200 **↗** SiO_2_. It is in perfect agreement with the findings of the textural analysis and SEM observations. The second main outcome is linked to the nature of the tested ATBs, as already noted in Figure 13. For each membrane, the ATBs slightly differ, following the parameters from Figure 10. SM and TMP displayed similar results (Table 10). For example, for the Mod_SWCNT-200 membrane, the diffusion coefficient is equal to 0.661 × 10^−10^ for SM and 1.1004 × 10^−10^ for TMP. The highest value of 1.8084 × 10^−10^ for TET could be, again, related to the higher reactivity of TET with the membrane, despite the larger molecule. The hypothesis is confirmed or even more pronounced with the incorporation of SiO_2_. 

### 3.5. Analysis of the Membranes by FTIR Spectroscopy

The attenuated total reflection (ATR) technique with the ZnSe crystal in the spectral range of 400–4000 cm^−1^ was used to inspect the functional groups present in the different membrane stages before and after oxidation, modification, and sorption experiment. The FTIR spectra were measured for each membrane stage. The Nicolet Avatar 360 spectrometer was set up for 128 scans for each sample measurement at room temperature. The collected spectra are shown in Figure 15. 

The FTIR spectra of each antibiotic (TET, TMP, and SM), the mildly oxidized SWCNT membranes (noted MO) before and after the sorption experiments of each ATB, and bare and modified SWCNT membranes are shown in Figure 15. As can be seen, SWCNT membranes (600; 400; 200) have similar bonds despite their state (bare vs. oxidized). The mildly oxidized SWCNT (SWCNT-MO) spectra highlight the prominent band of the carboxylic acid H bonded OH stretch around 3400 cm^−1^. Most probably, the formation of phenols during the oxidation took place, where the OH small band is observed in the same region as the OH stretch of the acid. The C-H stretch in the aromatic rings is observed around 3000 cm^−1^. In the 2000–1500 cm^−1^ region, several bands from overtones and combinations of substituted benzene rings stretch are observed. Additionally, around 1000 cm^−1^, the ring breathing mode and C-O vibrations were observed. The C-O bands in SWCNTs used in sorption are very intense and measured as the principal peak in all obtained spectra. The C-O bond represents the primary chemisorption measure of the ATB on the membranes. FTIR spectroscopy revealed the formation of a strong mono bond C-O during the chemisorption.

Moreover, this result indicates the probable degradation of the ATB during sorption. Other peaks on the used membrane were lower compared to the prominent one, except on the membrane used for SM sorption (SM/SWCNT-MO). We still witness SM characteristic peaks on the membrane, such as NH, NH_2,_ and SO_2_, at 1650 and 1200 cm^−1^, respectively. Furthermore, new vibration peaks for double-bonded C-O and C-N and C-S at 1700 and 1100 cm^−1^ appear on the spectra. The three peaks emphasize the degradation of SM in contact with SWCNT 0.2. The oxidized sites degrade the pollutant and the aromatic ring of the SWCNT, forming C=O, C=N, and C=S bonds with the pollutants. At 2990 and 1450 cm^−1^, we observe CH stretching, scissor bending, and antisymmetric deformation, respectively, of aliphatic compounds. At 1300 and 690 cm^−1^ are bands for the C-O and C-F stretch, which are characteristic of the aliphatic fluoro compound.

### 3.6. Analysis of the Membranes by XPS

A Kratos ESCA 3400 instrument equipped with a polychromatic Mg X-ray source of Mg Kα radiation (1253.4 eV energy) was used to determine the exact bonding involved in the sorption and to characterize the membrane surfaces. X-ray photoelectron spectra of the fresh membrane and the one used in SM sorption were determined. A wide scan range collected from 0–900 eV was measured on each sample. The scanning pressure was kept at around 10^−7^ Pa during the sample measurements.

The XPS spectra of SM, oxidized (SWCNT-MO), and used membranes for the sorption experiment (SWCNT-MO/SM) are shown in Figure 16. XPS analysis helps to see the changes in the surface. Boháčová et al., reported XPS analysis of unoxidized SWCNT membrane for effective H_2_/CO_2_ separation [44]. The SWCNT-MO spectra displayed a high intensity of carbon atoms in a region of 290–280 eV. Deconvolution of the carbon peak revealed oxidation bonds of the carbon C=O, C-O, and -COO at 287.2, 286.6, and 290.1 eV. Carbon rings and a small peak of the aliphatic chain are visible around 284.7 and 285.2 eV. The oxygen peaks confirmed the successful oxidation of the membranes. Peaks at 532 and 533.5 eV confirmed the persistence of the C=O and C-O bonds, respectively. A small peak of iron is also noticeable in the 700 eV region. The spectra of the used membrane SWCNT-MO/SM contain both characteristic peaks of SWCNT-MO and SM. Peak deconvolution of the SWCNT-MO/SM spectra revealed bond changes on the membrane surface. Sulfur peaks showed the appearance of sulfite groups at 164.6 eV adjacent to the sulfone groups of the adsorbed SM. Nearby to the oxygen region (530 eV), the C-O peak is present, and the C-O-C peak at 534 eV increases compared to the unused membranes; this is mainly due to the chemisorption reaction between the SWCNT functionalized reactive sites and SM. The effect was enforced on the carbon side (290 eV). There, O-COO is present, and the C-O peak rises at 288 and 287 eV correspondingly. The carbon–nitrogen monobond is visible at 400 eV, proving that even carbon rings contribute to the SM chemisorption. The diminution of the Fe peak at 710 eV is probably due to the deactivation of the active site or an overlaying of the sorbed molecules. The small Si peak at 100 eV is present, possibly from the SM impurities and the dissolution process. 

## 4. Discussion

The evolution of the appropriate hydrophobic SWCNT membranes required several steps:

200, 400, and 600 mg of SWCNT was added before milling in a preparation procedure of the membranes. The membranes diverged in thickness, consequently influencing the flow rate, the contact time, and the maximal sorption capacity of the membranes as a function of the pressure or the time. The amount of 200 mg was sufficient and optimal for the subsequent experiments. 

Only the highest amount of BP (300 mg) improved the sorption process. Otherwise, it slowed down the tests and was excluded from the next steps. However, the addition of SiO_2_ allowed a transparent study of how the porosity influences the separation processes. 

Mild oxidation of the SWCNT membranes increased the hydrophilicity and reactivity with the pollutants. It was found to be more suitable for the filtration or sorption process. Chemical oxidation of SWCNT was performed by a modified Hummers method using potassium permanganate in an acidic H_2_SO_4_/H_3_PO_4_ environment [44]. Compared to the classical Tour or Hummers oxidation procedure [54], a significantly smaller amount of permanganate was used approximately one-eighth of the original dose used in the Tour procedure). During such mild oxidation, the SWCNT underwent a significant structural rearrangement with a significant impact on its properties. The specified partial opening of the nanotubes and the introduction of oxygen-containing species reduced the number of free volume domains among the individual SWCNT fibers, reducing the overall permeability of the material but, on the other hand, improved mechanical properties [44]. The oxidized SWCNT form shows lower tensile strength but higher ductility. Such behavior indicates improved mechanical properties of oxidized samples that can undergo significant plastic deformation before the rupture [44]. Furthermore, the oxidation residuals, i.e., oxygen-containing functional groups (hydroxyl, epoxy, carbonyl, and carboxyl groups), have a significant impact on the properties of the material in terms of surface phenomena. In addition, the nature of groups can passivate or enhance specific molecular interactions of penetrants with oxidized SWCNT surface [55]. Passing species are either attracted and bound (e.g., van der Waals or hydrogen bridges) or repelled or blocked.

Non-oxidized samples were more appropriate for PT. Here, modification by the hydrophobic layer was needed. 

Together, the water contact angle, XPS, SEM, and textural analysis gave a unified picture of the prepared membranes. The highly porous nature of the materials containing SiO_2_ was confirmed, with high oxygen and low carbon content. The unmodified membranes, SWCNT-200, SWCNT-200 **↘** SiO_2_, and SWCNT-200 **↗** SiO_2_, revealed a higher magnitude of the specific surface area (~580 m^2^/g), the specific surface area of the mesopores (~350 m^2^/g), as well as the micropore volume (~110 mm^3^/g) than the modified membranes (~130 m^2^/g, 50 m^2^/g, and 30 mm^3^/g, respectively). Their total pore volume rises with an increasing amount of SiO_2_ (from 743 to 1218 mm^3^/g). On the other hand, the *S_BET_* of the modified membranes decreases with an increasing SiO_2_ content (from 172 to 117 m^2^/g). Even though the hydrophobic layer did not form a uniform layer, it completely covered the CNTs and SiO_2_ nanoparticles and provided a significantly different membrane surface for interaction with the ATBs. The combination of the hydrophilic CNTs and the hydrophobic grafted fluorinated polymer revealed no puff-like structure, as determined by SEM.

Mildly oxidized SWCNT membranes with or without BP adsorbed the entire amount of each ATB (TET, TMP, and SM) in less than 8 days. The MO membranes, differing in the amount of CNT and, thus, in thickness (from 118 to 267 µm), did not show any essential variations in the sorption process; all three sorption materials eliminated the amount of ATBs in less than 5 h. The same results were obtained in the treatment of the mixture of the three ATBs. However, the saturation tests showed that the maximal amount of ATBs for each membrane varies, linked to the diameter of the spherical enclosure and influenced by the polar surface area.

The time (2.3 s) in the thinnest membrane (118 µm) was sufficient for filtration of 98.8% of SM, 95.5% of TMP, and 87.0% of TET. The thicker membranes demonstrate a higher adsorption capacity, while the thinner ones exhibit much higher flow rates.

The increasing amount of SiO_2_ in the modified hydrophobic membranes enhanced the removal of the ATBs by pertraction. Similar to filtration, TET is more retained by the membranes than SM and TMP (the polar surface area is much higher than for the other two molecules), and it is more significant than SM or TMP. The diffusion coefficient of the ATBs varies between 0.7–2.7 × 10^−10^ and follows the order: Mod_SWCNT-200 < Mod_SWCNT-200 ↘ SiO_2_ < Mod_SWCNT-200 **↗** SiO_2_, which is in perfect agreement with the findings of the textural analysis and SEM observations. The highest values for TET could be, again, related to the higher affinity of TET to the membrane, despite it being the larger molecule. The hypothesis is confirmed with the incorporation of SiO_2_. 

The development of the appropriate membrane in this research area is still needed. Loose nanofiltration (NF) membranes coated by hydrophilic poly(dopamine) created via modification by gallic acid and polyethyleneimine were already used to remove ATBs from reclaimed wastewater [56]. However, NF requires high energy and removes essential minerals from the water, such as magnesium. Liu et al. (2017) examined the treatment of TET polluted water by several carbon-based media for membranes and found that graphene oxide with activated carbon was the most efficient membrane for TET removal [57]. Nevertheless, adsorbent membranes, which include carbon-based materials with high adsorption, pose problems of overloading and the need for regeneration. NF membrane was prepared also by molecular-layer-by-layer approach, using alternatively charged polyelectrolytes to remove micropollutants from saline reclaimed wastewater, including the ATBs amoxicillin and TET [58]. The membrane showed high rejection of the ATBs, while maintaining ~80% passage of salts including calcium chloride and sodium chloride. However, a problem in its robustness arises due to the use of a multilayered membrane with no covalent crosslinking.

## 5. Conclusions

New hydrophilic and modified hydrophobic SWCNT membranes were prepared and progressively applied in sorption, filtration, and pertraction experiments with the aim of eliminating three antibiotics, TET, SM, and TMP, as single pollutants or as a mixture. The amount of 200 mg of SWCNTs was sufficient and optimal for the described membrane experiments. BP slowed down the tests, but the addition of SiO_2_ allowed a transparent study of how the porosity influences the separation processes. Mild oxidation of the SWCNT membranes increased the hydrophilicity and reactivity with the pollutants during the filtration or sorption process. However, non-oxidised samples with a hydrophobic layer were more appropriate for pertraction. The total pore volume increased with an increasing amount of SiO_2_ (from 743 to 1218 mm^3^/g) in the hydrophilic membranes. The hydrophobic layer completely covered the CNTs and SiO_2_ nanoparticles, thus providing a significantly different membrane surface for interaction with the ATBs, revealing no puff-like structure in the SEM images. As sorption materials, CNTs eliminated the initial amount of ATBs in less than 5 h. The maximal amount of ATBs for each membrane could be linked to the diameter of the spherical enclosure and influenced by the polar surface area. A time of 2.3 s in the thinnest membrane (118 µm) was sufficient for filtration of 98.8% of SM, 95.5% of TMP, and 87.0% of TET; the thicker membranes demonstrate a higher adsorption capacity. The increasing amount of SiO_2_ in the modified hydrophobic membranes enhanced the removal of the ATBs by pertraction. The process is slower than filtration. However, it can lead to total elimination (e.g., 3 days for TET). Similar to filtration, TET is more retained by the membranes than SM and TMP. The diffusion coefficient of the ATBs varies between 0.7–2.7 × 10^−10^ and follows the order: Mod_SWCNT-200 < Mod_SWCNT-200 ↘ SiO_2_ < Mod_SWCNT-200 **↗** SiO_2_, which is in perfect agreement with the findings of the textural analysis and SEM observations. 

## Figures and Tables

**Figure 1 membranes-11-00720-f001:**
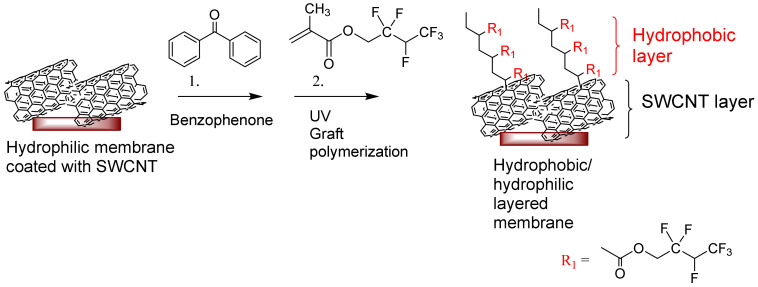
The graft-polymerization results in a hydrophobic layer on top of the SWCNT layer.

**Figure 2 membranes-11-00720-f002:**
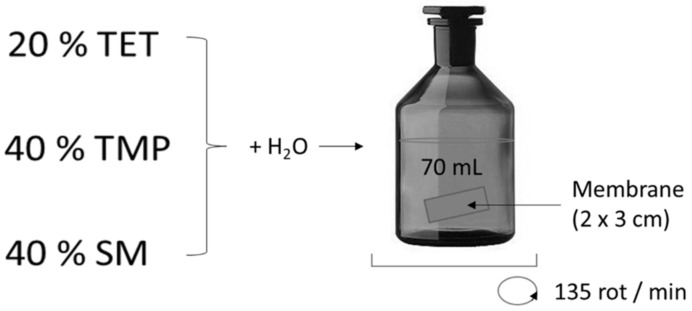
The membrane sorption of the antibiotics—tetracycline (TET), sulfamethoxazole (SM), and trimethoprim (TMP).

**Figure 3 membranes-11-00720-f003:**
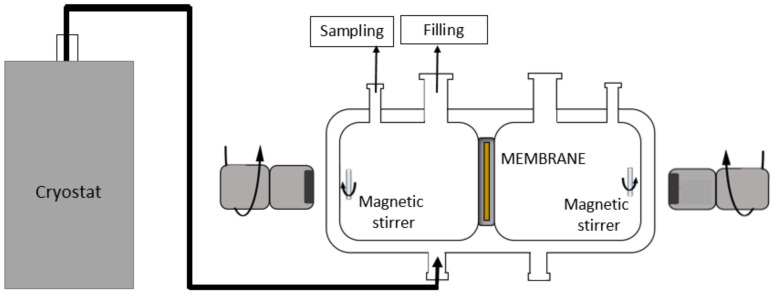
The scheme of the pertraction setup [48].

**Figure 4 membranes-11-00720-f004:**
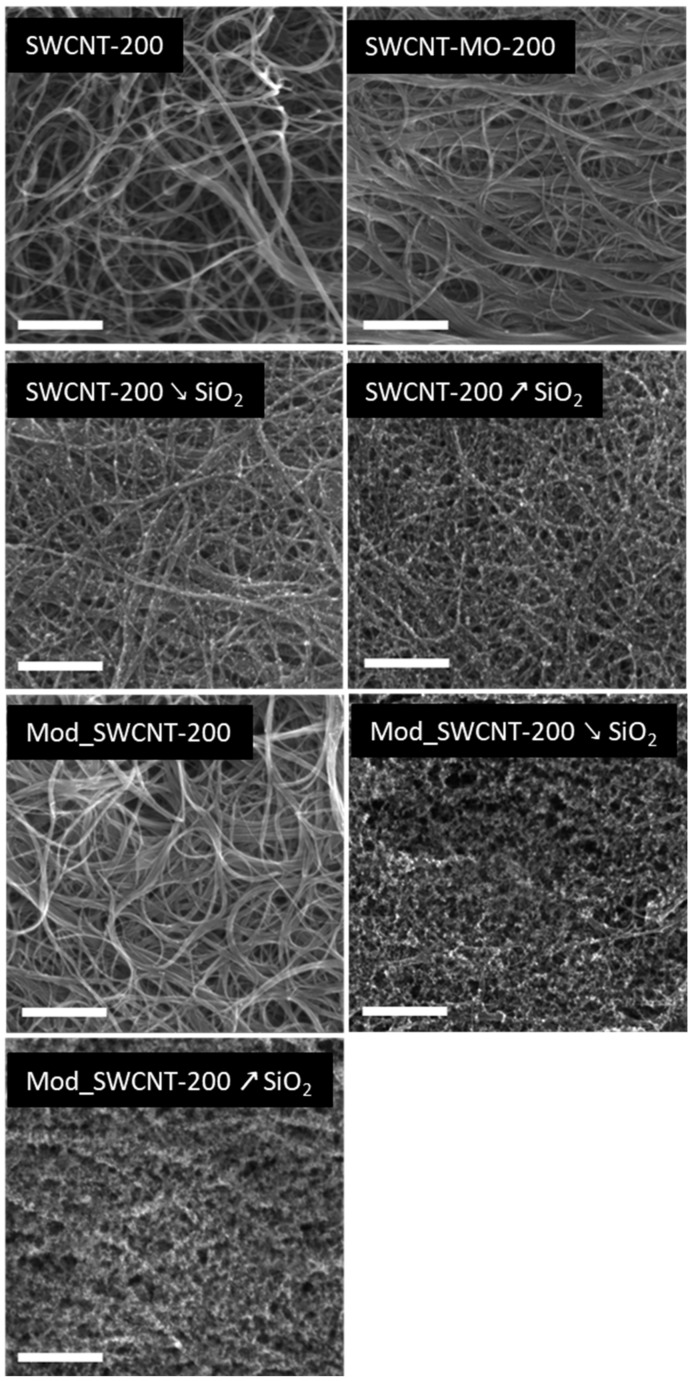
SEM images of the membrane surfaces. Scale bar corresponds to 500 nm.

**Figure 5 membranes-11-00720-f005:**
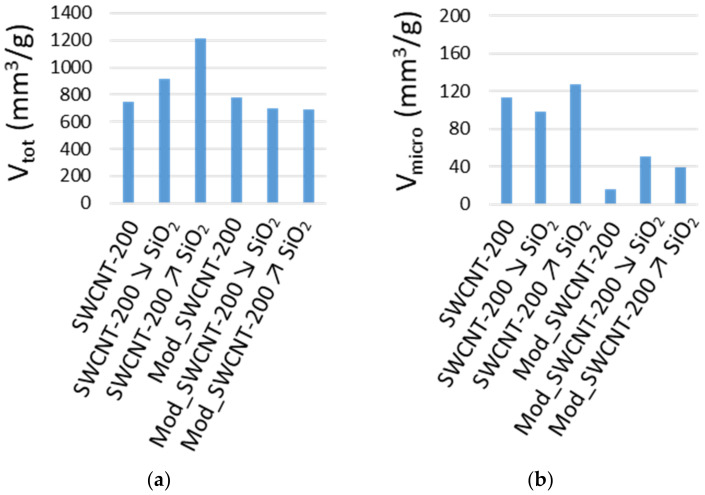
(**a**) Total pore volume; (**b**) micropore volume, determined from the N_2_ adsorption isotherms.

**Figure 6 membranes-11-00720-f006:**
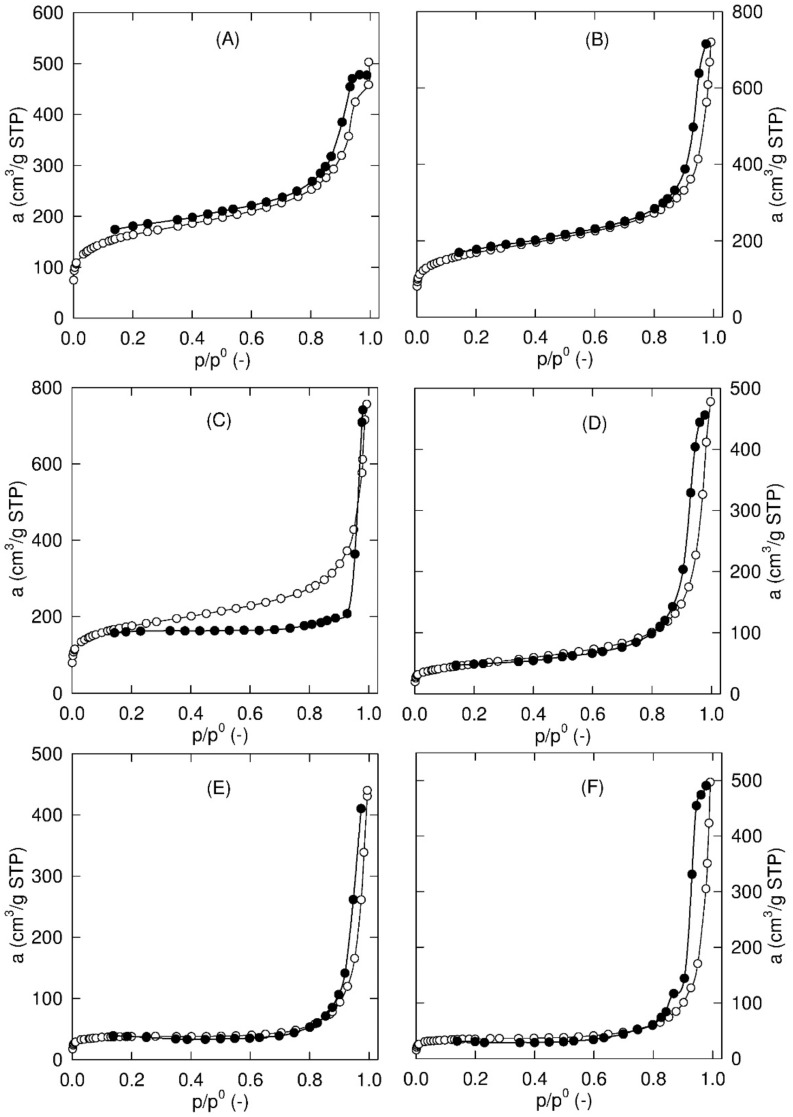
Adsorption (○) and desorption (●) branch of the nitrogen sorption isotherm for SWCNT-200 (**A**), SWCNT-200 **↘** SiO_2_ (**B**), SWCNT-200 **↗** SiO_2_ (**C**), Mod_SWCNT-200 (**D**), Mod_SWCNT-200 **↘** SiO_2_ (**E**), and Mod_SWCNT-200 **↗** SiO_2_ (**F**).

**Figure 7 membranes-11-00720-f007:**
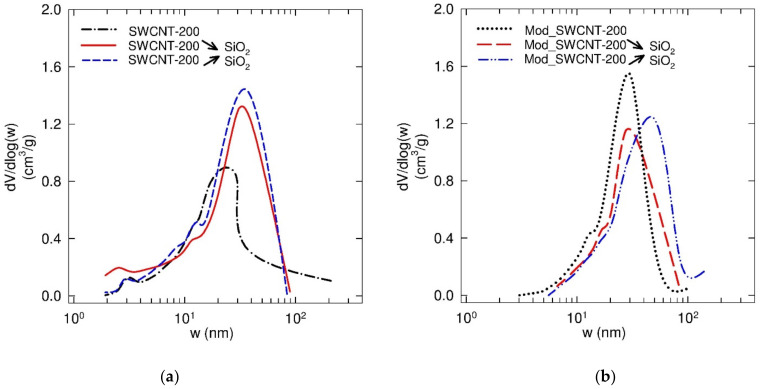
Pore size distribution evaluated from the desorption branch of the nitrogen sorption isotherm at 77 K: for unmodified (**a**) and modified (**b**) membranes by a hydrophobic layer.

**Figure 8 membranes-11-00720-f008:**
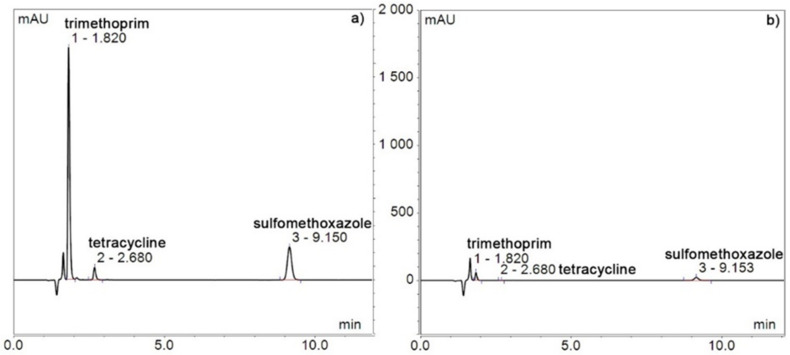
HPLC spectra of a mixture of ATB at time zero (**a**), a mixture of ATB after 1 day (**b**), and the SWCNT-MO-200 membrane.

**Figure 9 membranes-11-00720-f009:**
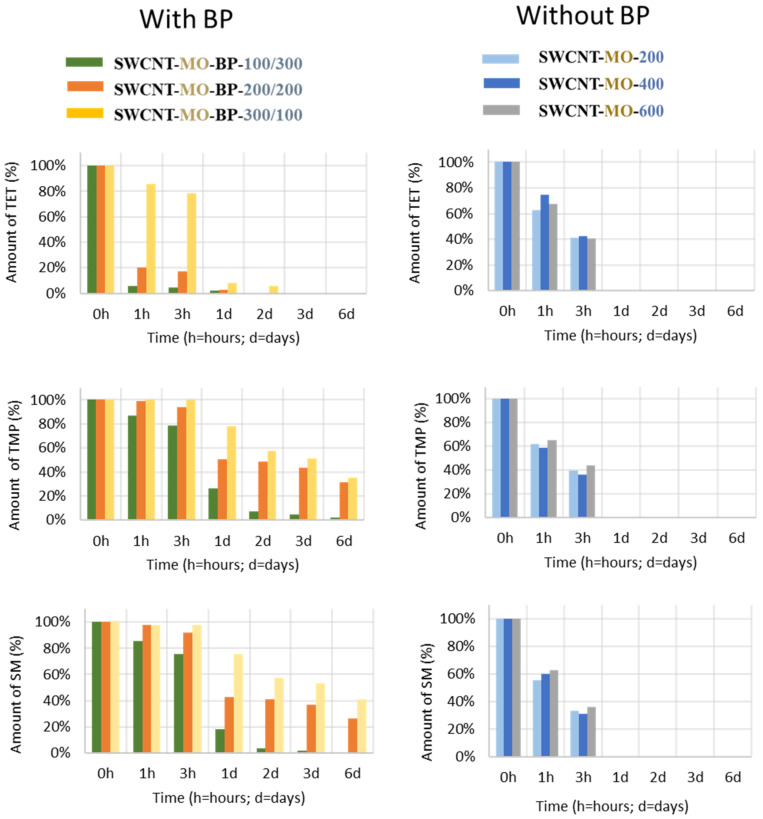
The evolution of ATB absorbed during the sorption experiments with SWCNT membranes.

**Figure 10 membranes-11-00720-f010:**
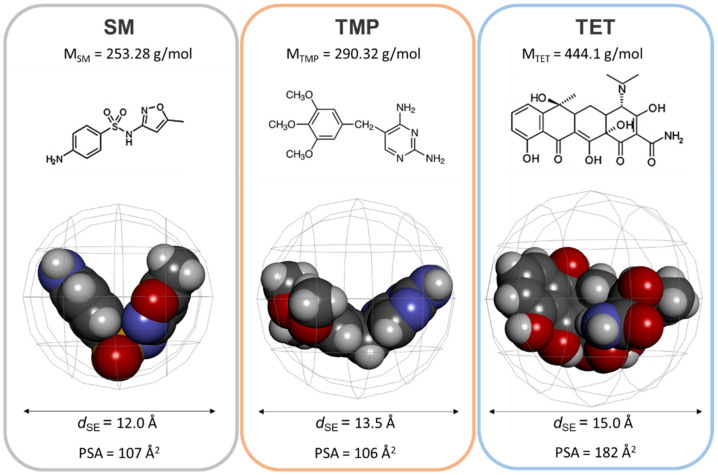
A comparison of the molecular weight (M_ATB_ in g/mol), structural formula, diameter of the spherical enclosure (*d*_SE_ in Å), and polar surface area (PSA in Å^2^) for SM, TMP, and TET.

**Figure 11 membranes-11-00720-f011:**
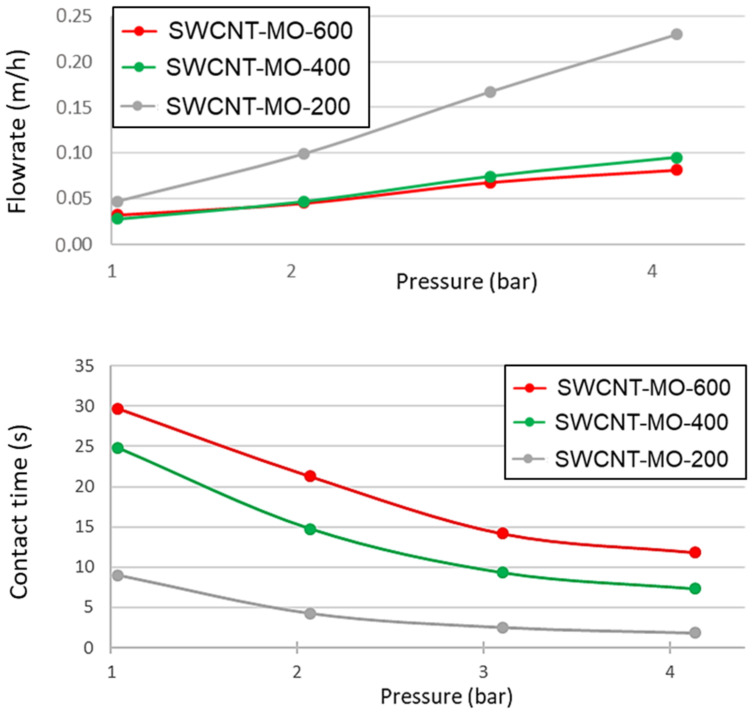
Flow rate and contact time of SWCNT-MO-200, SWCNT-MO-400, and SWCNT-MO-600 membranes as a function of the pressure.

**Figure 12 membranes-11-00720-f012:**
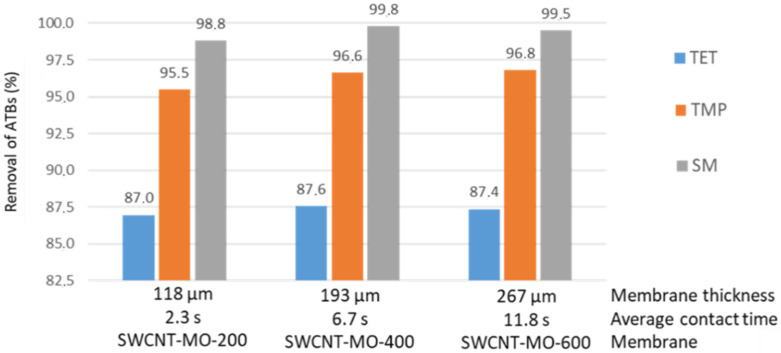
Rate of removal of ATBs for different membrane thicknesses versus average contact time.

**Figure 13 membranes-11-00720-f013:**
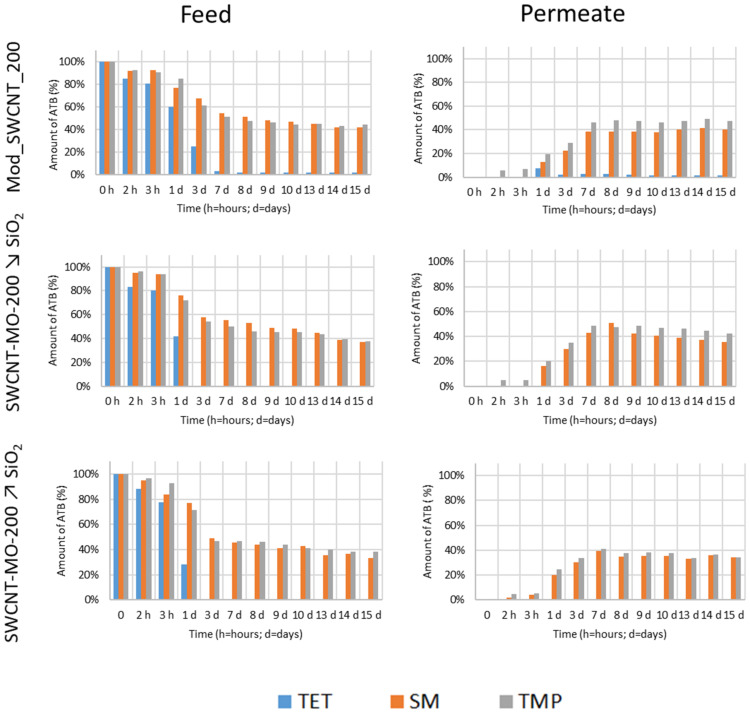
The amount of ATBs (%) in the feed and in the permeate for the modified membranes as a function of time.

**Figure 14 membranes-11-00720-f014:**
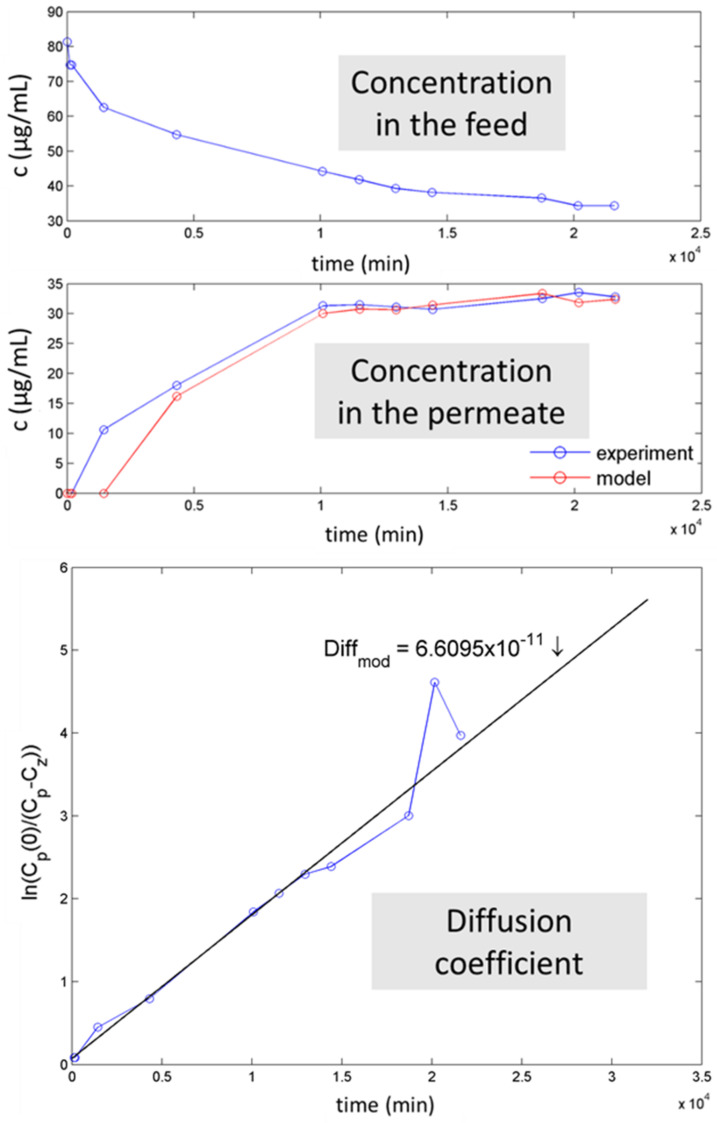
Concentration in the feed and permeate in time and an example of the calculation of the diffusion coefficient.

**Figure 15 membranes-11-00720-f015:**
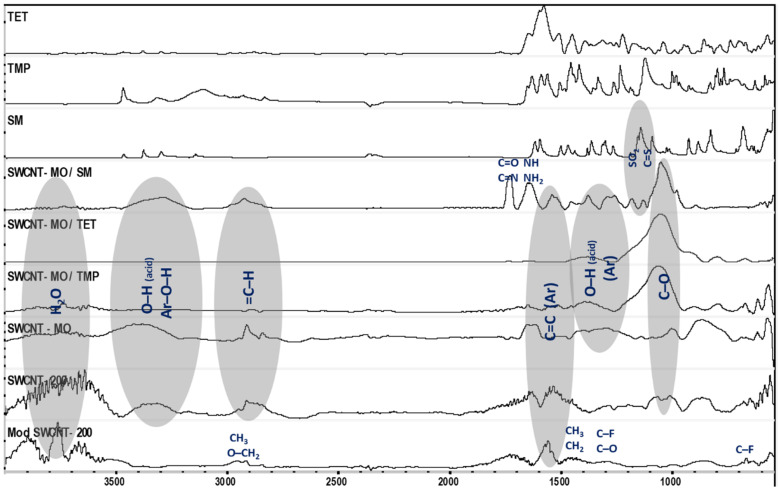
FTIR spectra of TET, TMP, SM, and selected membranes.

**Figure 16 membranes-11-00720-f016:**
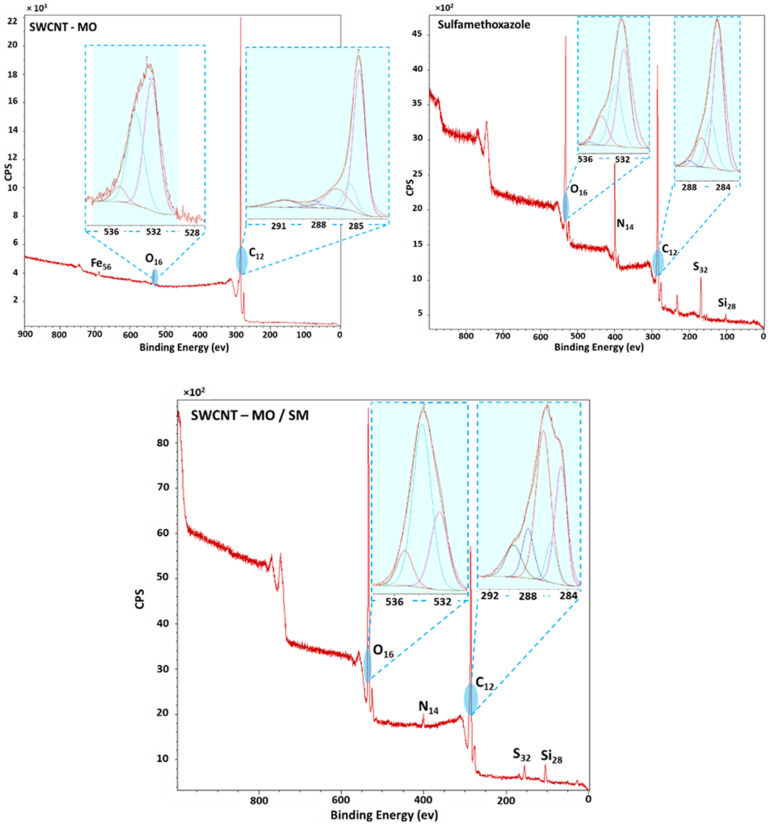
XPS spectra of the fresh and the used membrane SWCNT-MO-200 for SM sorption.

**Table 1 membranes-11-00720-t001:** Prepared SWCNT membranes.

Membrane Annotation	Weight of Dry SWCNT (or BP) Used for Membrane Preparation (mg)	Membrane Thickness (µm)
SWCNT-200	200	237
SWCNT-400	400	307
SWCNT-600	600	385

**Table 2 membranes-11-00720-t002:** Prepared mildly oxidized SWCNT membranes.

Membrane Annotation	Weight of Dry SWCNT (or BP) Used for Membrane Preparation (mg)	Membrane Thickness (µm)
SWCNT-MO-200	200	118
SWCNT-MO-400	400	193
SWCNT-MO-600	600	267
SWCNT-MO-BP-100/300	100–300	125
SWCNT-MO-BP-200/200	200–200	156
SWCNT-MO-BP-300/100	300–100	190

**Table 3 membranes-11-00720-t003:** SWCNT membranes enriched by SiO_2_.

Membrane Annotation	Weight of Dry SWCNT (or BP) Used for Membrane Preparation (mg)	Membrane Thickness (µm)
SWCNT-200 ↘ SiO_2_	200	202
SWCNT-200 ↗ SiO_2_	200	132

**Table 4 membranes-11-00720-t004:** SWCNT membranes with a hydrophobic layer.

Membrane Annotation	Weight of Dry SWCNT (or BP) Used for Membrane Preparation (mg)	Membrane Thickness (µm)
Mod_SWCNT-200	200	237
Mod_SWCNT-200 ↘ SiO_2_	200	202
Mod_SWCNT-200 ↗ SiO_2_	200	132

**Table 5 membranes-11-00720-t005:** Water contact angle and XPS results of unmodified and modified SWCNT membranes.

Membrane Type	Water-Drop Contact Angle	Elemental Composition by XPS			
		C1s	O1s	Si2p	F1s
SWCNT-200	63.5 ± 3.5°	95.25	4.75	BDL *	BDL *
Mod_SWCNT-200	80.1 ± 4.4°	94.07	5.55	BDL *	0.39
SWCNT-200 ↘ SiO_2_	N.D. *	6.89	59.67	33.44	BDL *
Mod_SWCNT-200 ↘ SiO_2_	N.D. *	7.71	58.99	32.73	0.58
SWCNT-200 ↗ SiO_2_	N.D. *	7.91	59.6	32.49	BDL *
Mod_SWCNT-200 ↗ SiO_2_	N.D. *	24.99	47.63	25.41	1.97

* N.D. = not determined; BDL = below detection limit.

**Table 6 membranes-11-00720-t006:** Elemental composition of prepared membranes obtained by SEM–EDS.

Sample	C at.%	O at.%	F at.%	Fe at.%	Si at.%	S at.%
SWCNT-200	89.95	6.68	-	2.57	0.63	0.17
SWCNT-200 ↘ SiO_2_	84.68	11.35	-	1.96	1.85	0.17
SWCNT-200 ↗ SiO_2_	72.41	19.85	-	1.88	5.86	-
Mod_SWCNT-200	94.50	4.80	-	0.30	0.10	0.30
Mod_SWCNT-200 ↘ SiO_2_	80.80	13.20	1.4	0.30	3.90	0.30
Mod_SWCNT-200 ↗ SiO_2_	59.10	28.00	1.3	0.20	11.2	0.20

**Table 7 membranes-11-00720-t007:** Results of textural assessments.

Sample	*S_BET_* (m^2^/g)	*S_BET_* (Kr) (m^2^/g)	*S_meso_* (m^2^/g)	*V_tot_* (mm^3^/g)	*V_micro_* (mm^3^/g)
SWCNT-MO-BP-400	–	0.14	–	–	–
SWCNT-200	561	–	351	743	113
SWCNT-200 ↘ SiO_2_	583	–	378	919	98
SWCNT-200 ↗ SiO_2_	579	–	340	1218	128
Mod_SWCNT-200	172	–	145	775	16
Mod_SWCNT-200 ↘ SiO_2_	130	–	24	698	50
Mod_SWCNT-200 ↗ SiO_2_	117	–	41	686	39

*S_BET_*: Specific surface area determined from the N_2_ adsorption isotherm at 77 K by the BET method. *S_BET_* (Kr): Specific surface area determined from the Kr adsorption isotherm at 77 K by the BET method. *S_meso_*: Specific surface area of mesopores evaluated by the *t*-plot method based on STSA equation suggested in the ASTM standard D-6556-01 (from N_2_ adsorption isotherm). *V_tot_*: Total specific pore volume evaluated from the N_2_ adsorption isotherm at *P*/*P_0_* = 0.99. *V_micro_*: Specific volume of micropores evaluated by the *t*-plot method based on STSA equation suggested in the ASTM standard D-6556-01 (from N_2_ adsorption isotherm).

**Table 8 membranes-11-00720-t008:** Sorption tests of ATBs using SWCNT membranes.

Test	Membrane	ATBs	c_ATBs_ (mg L^−1^)	Weight (mg)	Thickness (µm)	Active Surface (cm^2^)	Temperature (°C)
1	SWCNT-MO-200	SM	200	31	118	7.07	ambient
2		TMP					
3		TET					
4		MIX (SM/TMP/TET)	80/80/40				
5	SWCNT-MO-400	SM	200	62	193		
6		TMP					
7		TET					
8		MIX (SM/TMP/TET)	80/80/40				
9	SWCNT-MO-600	SM	200	90	267		
10		TMP					
11		TET					
12		MIX (SM/TMP/TET)	80/80/40				
13	SWCNT-MO-BP-100/300	SM	200	N/A	N/A		
14		TMP					
15		TET					
16		MIX (SM/TMP/TET)	80/80/40				
17	SWCNT-MO-BP-200/200	SM	200				
18		TMP					
19		TET					
20		MIX (SM/TMP/TET)	80/80/40				
21	SWCNT-MO-BP-300/100	SM	200				
22		TMP					
23		TET					
24		MIX (SM/TMP/TET)	80/80/40				

**Table 9 membranes-11-00720-t009:** Weight of adsorbed ATBs (SM, TMP, and TET) and amount of substance per area of the membrane after saturation of the SWCNT membranes.

Membrane	m_Adsorbed ATB_/*S*_membrane_ (mg/cm^2^)			n_Adsorbed ATB_/*S*_membrane_ (mmol/cm^2^)		
	SM	TMP	TET	SM	TMP	TET
SWCNT-MO-200	1.5	2.0	6.0	5.9	6.9	13.5
SWCNT-MO-400	3.0	3.5	8.5	11.8	12.0	19.1
SWCNT-MO-600	4.0	4.5	10.0	15.8	15.5	22.5

**Table 10 membranes-11-00720-t010:** Diffusion coefficients of SM, TMP, and TET transport through the selected membranes.

Membrane	Diffusion Coefficient *D* (m s^−1^) × 10^−10^		
	SM	TMP	TET
Mod_SWCNT-200	0.6609	1.1004	1.8084
Mod_SWCNT-200 ↘ SiO_2_	1.3871	1.4149	N.D. *
Mod_SWCNT-200 ↗ SiO_2_	2.5986	2.7443	N.D. *

* N.D. = Not determined.

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
