# Peer review of "Modified Single-Walled Carbon Nanotube Membranes for the Elimination of Antibiotics from Water"

_membranes, 2021, doi:10.3390/membranes11090720_

Round 1

Reviewer 1 Report

  1. You do not need to illustrate very detailed information of ATBs’ functions or undesired effects for human beings from lines 55 to 63. Instead, you can say the production amount or usage amount in the world.
  2. Line 124, why does this study focuses on SWCNTs instead of MWSNTs and MWCNT-COOH?
  3. Could you please explain the Table 2 weight before milling (mg) 100-300, 200-200, and 300-100? Maybe it’s better to add some notifications under the table.
  4. Table 1, 2, 3, and 4. Does the membrane thickness has been analyzed by statistics (standard deviation)?
  5. Could you add some illustrations about Table 6 in the manuscript? Cuz the element composition is related to the adsorption ability. In addition, what is F? Are there any other elements in the membrane?
  6. Figure 5, remove the unit to the ordinate.
  7. Figure 7, mark the curves as symbols and lines in case they cannot be read once it is printed as black and white color.
  8. Line 451, Chyba! Nenalezen zdroj odkazů.8 is Table 8?
  9. Figure 9 is difficult to read unless you marked each figure and change the color for SWCNT-MO-BP or SWCNT-MO-200.

Author Response

Author's Reply to the Review Report (Reviewer 1) has been attached in the file "Rev. 1"

Reviewer 2 Report

The paper contains many abbreviations that are presented both in the article and at the end of the article to facilitate the work. However, there are some abbreviations for which it was not specified what they represent, for example: CNB (page 3 lines 118 and 119 and page 10 line 377) or NA page 2 line 71. 

There are many typos that need to be corrected. Example: “Chyba! Nenalezen zdroj odkazů.”

In Chapters 3 or 4, I recommend a comparison of the results obtained with those obtained by other research teams for other types of membranes for the elimination of antibiotics from water.

Author Response

Author's Reply to the Review Report (Reviewer 2) has been attached in the file "Rev. 2"

Reviewer 3 Report

The manuscript number, titled “Modified single-walled carbon nanotube membranes for the elimination of antibiotics from water”. The manuscript is well-presented. However, there are some minor issues to be addressed before publication. I, therefore, recommend to be published after minor revision.

  1. The authors should arrange the pore-size distribution curves of the individual materials along with N2 adsorption-desorption isotherms which would be easy for the readers
  2. Be consistent while writing the single-walled carbon nanotubes/SWCNT throughout the manuscript.
  3. The authors should provide the XPS spectra of SWCNT before and after oxidation.
  4. The authors should provide the oxidation mechanism.
  5. Perfluorinated methacrylate should be perfluorinated (meth)acrylate.
  6. It is confusing the presence of Fe and S (SEM-EDS) in the prepared samples.

Author Response

Author's Reply to the Review Report (Reviewer 3) has been attached in the file "Rev. 3"
